# Structures of peptide-free and partially loaded MHC class I molecules reveal mechanisms of peptide selection

Raghavendra Anjanappa[1,7], Maria Garcia-Alai[2,7], Janine-Denise Kopicki[3], Julia Lockhauserbäumer[3], Mohamed Aboelmagd[1], Janina Hinrichs[2], Ioana Maria Nemtanu[2], Charlotte Uetrecht[3,4], Martin Zacharias[5], Sebastian Springer[1✉] & Rob Meijers[2,6✉]

Major Histocompatibility Complex (MHC) class I molecules selectively bind peptides for presentation to cytotoxic T cells. The peptide-free state of these molecules is not well understood. Here, we characterize a disulfide-stabilized version of the human class I molecule HLA-A*02:01 that is stable in the absence of peptide and can readily exchange cognate peptides. We present X-ray crystal structures of the peptide-free state of HLA-A*02:01, together with structures that have dipeptides bound in the A and F pockets. These structural snapshots reveal that the amino acid side chains lining the binding pockets switch in a coordinated fashion between a peptide-free unlocked state and a peptide-bound locked state. Molecular dynamics simulations suggest that the opening and closing of the F pocket affects peptide ligand conformations in adjacent binding pockets. We propose that peptide binding is co-determined by synergy between the binding pockets of the MHC molecule.

[1] Department of Life Sciences and Chemistry, Jacobs University Bremen, Bremen, Germany. [2] European Molecular Biology Laboratory, Hamburg Outstation, Hamburg, Germany. [3] Heinrich Pette Institute, Leibniz Institute for Experimental Virology, Hamburg, Germany. [4] European XFEL GmbH, Schenefeld, Germany. [5] Physics Department, Technical University of Munich, Garching, Germany. [6] Present address: Institute for Protein Innovation, Boston, USA. [7] These authors contributed equally: Raghavendra Anjanappa, Maria Garcia-Alai. ✉email: s.springer@jacobs-university.de; rob.meijers@proteininnovation.org

Major histocompatibility complex (MHC) class I molecules bind peptides in the lumen of the endoplasmic reticulum and present tightly binding (high-affinity) peptides on the cell surface for recognition by cytotoxic T cells[1]. It is generally assumed that initially, just after folding of the heavy chain and its association with the light chain, beta-2 micro-globulin (β2m), the peptide-binding site is occupied by low-affinity peptides, followed by peptide exchange and quality control such that the class I molecule is associated with high-affinity peptides when it reaches the cell surface[2–4]. Such high-affinity peptides have a low dissociation rate, which depends on tight binding to defined pockets of the binding groove, among them the A pocket (which binds the amino terminus of the peptide) and the F pocket (which binds the carboxy-terminal side chain of the peptide)[5].

A thorough understanding of the selection of high-affinity peptides by class I molecules is essential for the design of peptide vaccines and for the analysis and prediction of cytotoxic T-cell responses to infection and cancer[6]. As peptide-free class I molecules could not be crystallized so far, they have been mod-eled in various ways[7,8]. In these modeling approaches, the con-formations of the class I side chains that line the peptide-binding groove are derived from peptide-loaded class I structures[9]. Thus, such models cannot claim to represent the true shape of the peptide-free binding groove at the point of peptide recognition.

To better understand the peptide-binding process to MHC class I, we have recently stabilized class I molecules by the introduction of a disulfide bond between the alpha helices in the vicinity of the F pocket[10]. Such disulfide-stabilized class I mole-cules can be folded in vitro with dipeptides to obtain recombinant soluble peptide-free MHC class I molecules that rapidly bind exogenous peptide[11]. Importantly, over a hundred complexes of the disulfide-stabilized form of the frequent human class I allo-type HLA-A*02:01 [HLA-A*02:01(Y84C/A139C), in the follow-ing named dsA2 with different peptides bind the T-cell receptor (TCR) with the same affinities as their wild-type A*02:01 (wtA2) counterparts[12]. The crystal structure of the complex of the 1G4 TCR and dsA2 with the cognate peptide shows that peptide binding and interaction with the TCR are not affected by the disulfide bond[13]. Thus, the dsA2 molecule appears as an ideal model system to study empty class I molecules.

In the work presented here, we prepare a peptide-free form of dsA2 and demonstrate the absence of peptide in the binding groove by mass spectrometry (MS) and differential scanning fluorimetry. Through several X-ray crystal structures, we show that peptide binding is accompanied by concerted conformational switches of the amino-acid side chains of A2 that line the A and F pockets of the binding groove. We use molecular dynamics (MD) simulations to show that the binding of the C-terminal peptide residue into the F pocket causes these side chain conformational changes that propagate to other pockets and influence the binding of other residues of the peptide.

## Results

**Disulfide-stabilized HLA-A*02:01 is stable without peptide**. We used high concentrations of dipeptides to refold bacterially expressed wtA2 and dsA2 from inclusion bodies together with β2m, but without full-length peptide[11], and we purified the complexes by size exclusion chromatography (SEC; Supple-mentary Fig. 1). To quantify the stabilization of dsA2 by the disulfide bond, we then performed thermal denaturation assays at different concentrations of free GM (glycyl-methionine) dipeptide. In these assays, the characteristic decrease of trypto-phan fluorescence upon protein unfolding is used to determine the transition midpoint of denaturation (melting temperature,

$T_m$) of the protein. The fluorescence vs. temperature curves for both wtA2 and dsA2 show two separate melting events, one at 62 °C for β2m and another at lower temperature for the heavy chain (Fig. 1). The melting event for the heavy chain is sensitive to the concentration of free GM, suggesting that GM binds to and stabilizes the peptide-binding site as predicted[11]. The dependence of the $T_m$ on dipeptide concentration matches our previous observation for low-affinity ligands[14]. Remarkably, the dsA2 heavy chain gave a clear transition signal even at low dipeptide concentrations (at < 5 mM), whereas wtA2 did not give a distinct transition under these conditions. We conclude that the Cys84–Cys139 disulfide bond of dsA2 enables it to remain folded even at low concentrations of dipeptide.

We then performed native MS[15] on dsA2 after removing any external dipeptide by SEC. Remarkably, the most-abundant species was the dsA2 complex (Fig. 2a, top). Upon activation in the mass spectrometer, individual heavy chain or β2m molecules are released from the complex and appear as individual peaks (Supplementary Fig. 2 and Supplementary Table 1). Importantly, we did not detect the dipeptide used for refolding either free or bound to dsA2, which confirms that the dsA2 complex is indeed empty and stable in absence of any peptide in the binding groove (Fig. 2a, top). However, the dipeptide was detected in the low $m/z$ when added to the solution. In contrast to dsA2, there were only minute amounts of folded wtA2 when dipeptide was absent (Fig. 2a, third panel). Still, wtA2 did remain stable when 0.5 mM free GM or GL (glycyl-leucine) were present (Fig. 2a, bottom). Thus, from the thermal denaturation and the MS experiments together, we conclude that although wtA2 is dependent on a high concentration of dipeptide to maintain its folded conformation, the dsA2 molecule is conformationally and thermally stable in the absence of any peptide or dipeptide in the binding groove.

We next investigated whether empty dsA2 was able to bind cognate peptides. Previously, we had shown that dipeptide complexes of wild-type class I molecules can exchange bound dipeptide for a full-length high-affinity peptide[11]. Therefore, we next asked whether dsA2 was still able to bind NLVPMVATV (NV9), a high-affinity nonamer peptide. A second nonapeptide (YPNVNIHNF, YF9) with low affinity was used as a control. We added a tenfold molar excess of NV9 to dsA2 and analyzed the resulting complexes by native MS (Fig. 2b). As predicted, dsA2/NV9 was by far the most-abundant species at low collision voltage. The NV9 peptide remained bound at higher activation up to 75 V. Likewise, measurements with substoichiometric peptide concentration (10:1) were performed to exclude the possibility of artefactual complex formation. Despite this low NV9 concentra-tion, dsA2/NV9 was detected by native MS, whereas the low-affinity control was not visible (Fig. 2c, second panel). In MS/MS experiments, the selected protein–peptide complex dissociated into the dsA2 heavy chain/β2m dimer (without the peptide) as well as into the individual subunits and the nonapeptide (Fig. 2c, third & fourth panel). We conclude that the binding of full-length peptides to dsA2 is not affected by the presence of the Cys84–Cys139 disulfide bond.

**Structure of dsA2 with dipeptides in the A and the F pockets**. To determine in what region of the dsA2-binding groove the dipeptides bind, we folded dsA2 with dipeptide (GL or GM) as above and crystallized the complex with 2 mM free dipeptide present. High-resolution crystal structures were obtained for dsA2/GL (at 1.6 Å) and dsA2/GM (at 1.7 Å, see Table 1 and Supplementary Table 2 for further details). The structures show one dipeptide in the A pocket and in addition one glycerol molecule from the buffer in the F pocket (Fig. 3a). In terms of its conformation, the GL dipeptide in the A pocket is identical to the

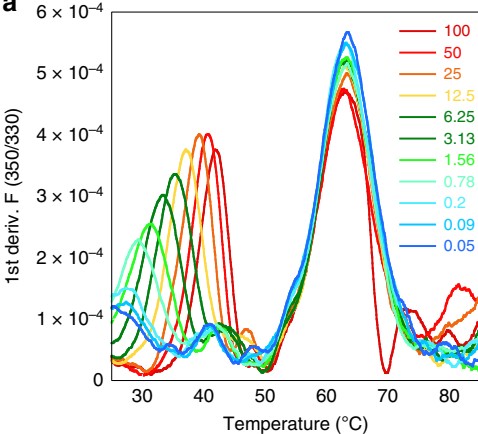

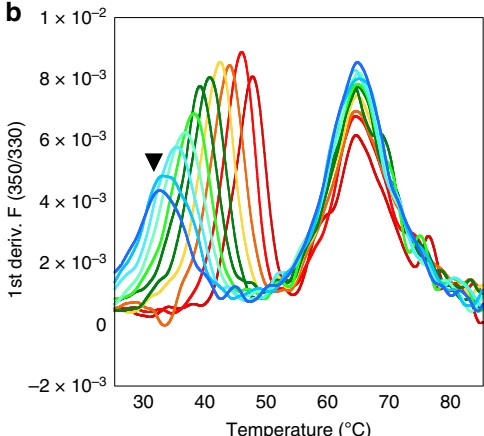

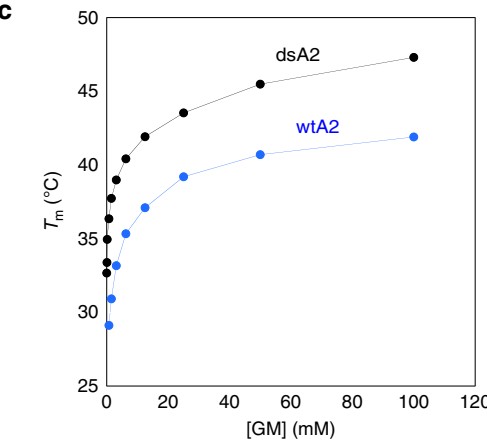

**Fig. 1 dsA2 is stable at very low dipeptide concentrations.** Thermal denaturation of A2 in the presence of increasing concentrations of GM measured by nanoDSF (tryptophan fluorescence). **a**, **b** First derivatives of the F350/F330 (ratio of fluorescence emissions at 350 and 330 330 nm) curves for wtA2 **a** and dsA2 **b** with the maxima reporting the melting temperatures ($T_m$). The concentration of GM (in mM) is indicated in rainbow colors from red (maximum, 100 mM) to blue (lowest, 0.05 mM). The arrow points to the transition of the dsA2 heavy chain that is visible even at very low dipeptide concentration. **c** Scatter plot of the $T_m$ obtained for the A2 heavy chain (first unfolding event in A) as a function of GM concentration (mMMsmall M).

first two residues of the peptide in the wtA2/GLCPLVAML crystal structure (PDB [protein data bank] code 3MRF). Thus, the A pocket of A2 can bind a dipeptide in the same manner as it binds the first two residues of a longer peptide.

As glycerol molecules tend to bind into the MHC class I peptide-binding groove and to stabilize the MHC fold in partially empty crystal structures[16], we switched to crystals of dsA2/GM that formed in the presence of 1,2-ethanediol (EDO) at 10 mM free dipeptide and determined a crystal structure of dsA2 at 1.6 Å (Fig. 3b). EDO, which is a cryo-protectant like glycerol, is smaller and easier to remove from the binding groove (see below). This crystal form is monoclinic P2₁, and it contains two molecules in the asymmetric unit. In the following, we describe only one molecule unless substantial differences occur between the two molecules.

In this structure (dsA2/GM₂), two GM dipeptides are present in the peptide-binding groove, one in the A pocket and one in the F pocket. The latter GM occupies positions P8 and P9 of a full-length nonamer peptide, and the carboxy (C) terminus of the GM superimposes well with the C terminus of previously determined structures with full-length peptides that have a C-terminal leucine (Fig. 3c, d). The conservation of the structure of the peptide C terminus in dsA2 was also previously noted in the dsA2/NV9 structure[13], which indicates that the Tyr84Cys mutation has little effect on the peptide conformation. The GM dipeptide in the F pocket has higher atomic displacement parameters (average for all atoms, 33 Å²) than the GM in the A pocket (25 Å²) and than the surrounding residues of dsA2 (25 Å²). This indicates that the GM in the F pocket is more mobile and possibly prone to leaving the peptide-binding groove. In this crystal, there is no other ligand in the peptide-binding groove apart from the two dipeptides, and the rest of the groove is filled with solvent molecules. We conclude from this structure that dsA2 can accommodate two dipeptides at the same time, one in the A and one in the F pocket, and that they are bound like the N and C termini of a nonamer peptide.

**Structures of a peptide-free dsA2 molecule.** We next wished to obtain a peptide-free structure of dsA2. As the peptide-free dsA2 characterized above did not yield crystals, we used dsA2/GM crystals and vacated the peptide-binding groove of the GM dipeptide. To do this, we first alternated between crystal washing in cryobuffer (lacking dipeptide) and crystal cryo-cooling to 100 K (an "annealing" protocol). After two cycles of washing and cryo-cooling, we obtained a crystal structure of dsA2 at 1.75 Å resolution that is devoid of any amino acids in the peptide-binding groove (dsA2/peptide_free-1, Fig. 4a, Table 1). There is one molecule of EDO in the A pocket (Fig. 4b), which mimics the N-terminal amino group and the peptide backbone conformation.

The lack of peptide in the A pocket of dsA2 is accompanied by an interesting conformational change of some side chains. In the empty structure, the side chain of Tyr99 on the β-sheet floor of the peptide-binding groove has changed its conformation to point further into the A pocket, away from His70 on the α₁ helix (Fig. 4b). There is no longer a hydrogen bond network connecting these residues, and the closest distance between His70 and Tyr99 is 4.3 Å. The side chain of His70 has moved out of the B pocket into the C pocket. It occupies an area that in peptide-bound structures is filled with the side chain of the P6 residue. The side chain of Phe9, which is also situated on the β sheet floor, has turned perpendicular to the peptide-loaded conformation, and now fills the space vacated by the side chain of His70. This side chain movement by Phe9 may sterically hinder the imidazole ring of His70 and contribute to the disruption of the His70-Tyr99 hydrogen bond. The A pocket has expanded as a consequence of the Tyr99 and His70 rearrangements and thus appears ready to accommodate an incoming peptide. The concerted movement of the Phe9-His70-Tyr99 triad arranges the A and the B pockets for

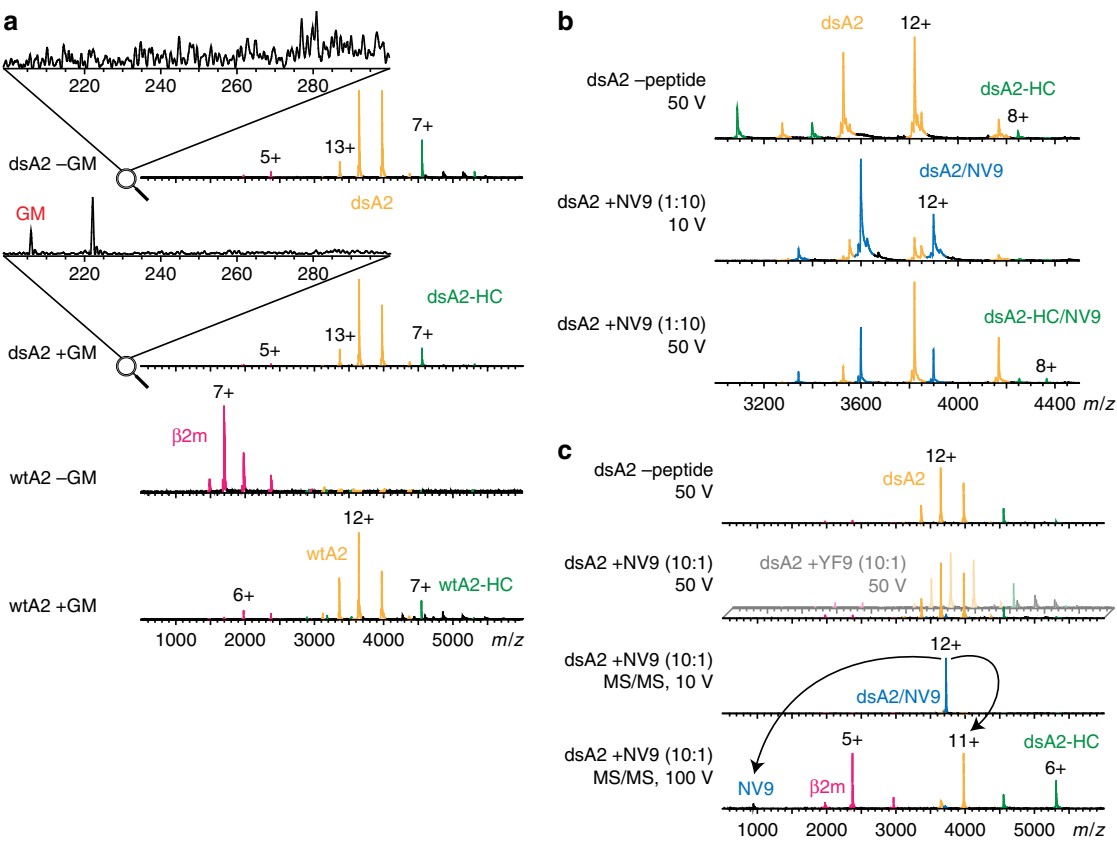

**Fig. 2 Stability and peptide binding of dsA2 measured by native MS. a** Representative mass spectra of dsA2 and wtA2, 10 μM, respectively, in 250 mM ammonium acetate, pH 8 recorded at a collision energy of 50 V. Pink, β2m; green, A2 heavy chain (A2-HC); yellow, A2 complex. Comparing the spectrum of dsA2 alone (top) with the spectrum of dsA2 in the presence of 0.5 mM GM (second panel), the low m/z range (zoom) clearly shows that the protein complex no longer contains any dipeptide used for refolding. Note, the dipeptide owing to low-affinity dissociates easily at the activation energies necessary to resolve the peaks sufficiently. The stability of dsA2 does not depend on the presence of dipeptide (top), whereas wtA2 without GM is hardly visible (third panel). In presence of 0.5 mM GM, the wtA2 complex can be detected in high abundance, although the wtA2/GM complex is not observed (bottom). **b** Superstoichiometric binding of NV9 high-affinity peptide to dsA2. Green, dsA2 heavy chain; yellow, dsA2 complex; blue, dsA2/NV9 complex. Mass spectrum of dsA2 at collision voltage of 50 V is given as a reference (top). At low collision voltage (10 V), the peptide-bound complex is the most-abundant species upon addition of 0.1 mM NV9 (middle). When increasing the voltage to 50 V, the complex begins to dissociate. However, some heavy chain/NV9 peptide complex can be detected (8+ peak), indicating high stability (bottom). **c** Substoichiometric binding of NV9 high-affinity peptide to dsA2. Pink, β2m; green, dsA2 heavy chain; yellow, dsA2 complex; blue, dsA2/NV9 complex. Reference mass spectrum of dsA2 at 50 V collisional energy (top). Peaks for dsA2/NV9 are gathered from a mixture of dsA2 and high-affinity nonapeptide NV9 (10:1), whereas the low-affinity control YF9 gives no signal (second panel). The 12+ peak at 3723 m/z was selected for MS/MS analysis at 10 V in the collision cell (third panel). The MS/MS spectrum measured at 100 V clearly shows the dissociation of the complex into dsA2 as well as its subunits and NV9 (bottom).

**Table 1 Crystal structures of dsA2 and wtA2 compared in this work.**

| Structure | Treatment | Groove occupancy | | PDB code | First described |
|---|---|---|---|---|---|
| | | **A pocket** | **F pocket** | | |
| dsA2/peptide_free-1 | Annealing | 1 EDO | 1 EDO | 6TDR | This work |
| dsA2/peptide_free-2 | High-salt treatment | 2 EDO | solvent | 6TDS | This work |
| dsA2/peptide_free-2[a] | High-salt treatment | 2 EDO | solvent | 6TDS | This work |
| dsA2/GM2 | EDO | 1 GM | 1 GM | 6TDQ | This work |
| dsA2/GM | Glycerol | 1 GM | 1 glycerol | 6TDO | This work |
| dsA2/GL | GL addition | 1 GL | 1 glycerol | 6TDP | This work |
| dsA2/NLVPMVATV | – | NLVPMVATV (entire groove) | | 6Q3K | Ref. [12] |
| wtA2/LGYGFNVNY | – | LGYGFNVNY | | 3PWL | Ref. [37] |
| wtA2/CLGGLLTMV | – | CLGGLLTMV | | 3REW | Ref. [38] |
| wtA2/GLCPLVAML | – | GLCPLVAML | | 3MRF | Unpublished |
| wtA2/UVcleaved | UV cleavage | KILG | Valine | 2X4N | Ref. [39] |

[a]Second molecule in the asymmetric unit.

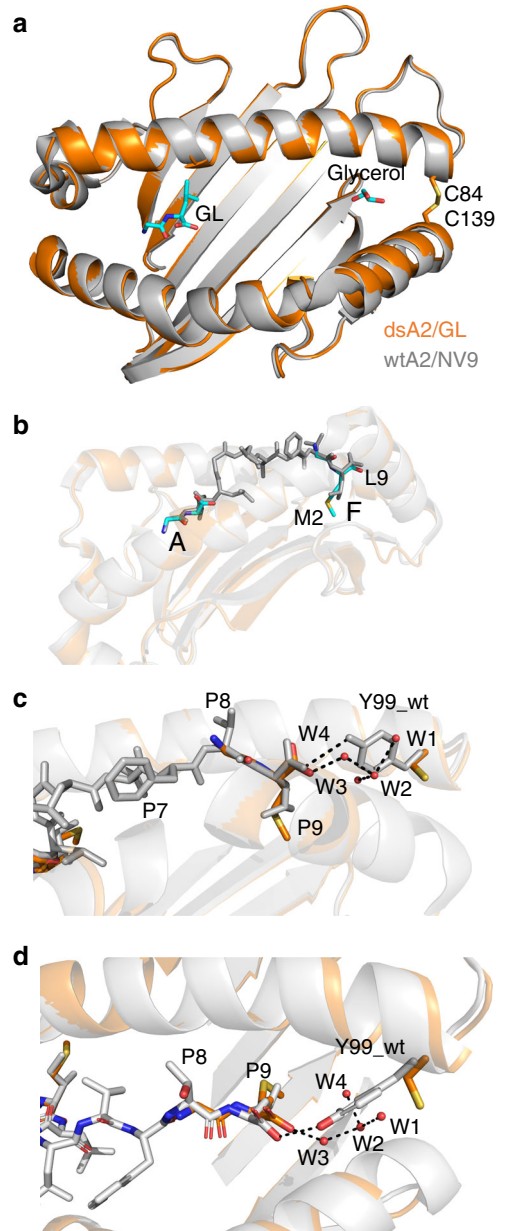

**Fig. 3 Crystal structures of dsA2 in complex with dipeptides in the A and F pockets. a** Overlay of the peptide-binding groove with a ribbon diagram of dsA2/GL in gold and wtA2/GLCPLVAML in gray. The C84-C139 disulfide bond is shown as sticks. **b** The dsA2/GM$_2$ structure (gold; see Supplementary Table 2) is shown overlaid with the published wtA2/GLCPLVAML structure (pdb 3MRF; gray). View from the side of the $\alpha_2$ helix with comparison of the GLCPLVAML peptide (gray) with the GM dipeptides (cyan) in the A and F pocket (peptides depicted as sticks). A transparent cartoon of the secondary structures of the peptide-binding groove is shown for orientation. **c, d** Zoom into the F pocket of the same structure, showing that the carboxylate group of the GM dipeptide is in the same orientation as the carboxyl group of the full-length peptide, despite the Tyr84Cys mutation. In dsA2/GM$_2$, water molecules (W1, W2, W3, and W4) fill the space where the side chain of Tyr84 is pointing in wtA2/GLCPLVAML.

peptide binding. We propose to call this peptide-receptive conformation the unlocked state of the A pocket, defined by the release of the hydrogen bond between His70 and Tyr99 and the upward movement of the side chain of Phe9.

We then obtained a second crystal structure of peptide-free dsA2 (determined at 1.70 Å) in a high-salt cryo-protection solution. In this structure (dsA2/peptide_free-2), two EDO molecules are present in the A pocket (Supplementary Fig. 3). Remarkably, the Phe9 side chain occurs in the unlocked conformation in one molecule of the asymmetric unit, and a conformation that corresponds to peptide-loaded A2 in the other molecule (which we will call the 'locked', or peptide-bound, state). This conformation seems to depend on the position of the EDO molecule and further suggests that Phe9 has a role in shaping the B pocket and in binding the side chain of the P2 residue of the peptide. Here, the His70 and Tyr99 side chains are within weak hydrogen bonding distance (3.40 Å) for the dsA2 molecule in the asymmetric unit that is in the locked conformation. In the other dsA2 molecule, the distance is 3.52 Å. In both peptide-binding grooves, the EDO molecules form hydrogen bonds with Tyr99 (Supplementary Table 3). From the peptide-free dsA2 structures, we conclude that the triad Phe9-His70-Tyr99 responds to peptide binding and accommodates the peptide side chains to switch from an unlocked to a locked conformation of the A pocket.

To assess the role of this allosteric Phe9-His70-Tyr99 triad in adapting the A pocket of A2 we verified whether these conformations exists among the published wtA2 structures. We analyzed the distribution of the hydrogen bond distance between the side chains of His70 and Tyr99 (Fig. 4c), and found >90% of these structures do not have such a hydrogen bond. Strikingly, out of the 258 peptide-loaded structures in the PDB, only three structures show a Phe9 side chain conformation that is similar to the dsA2/empty structures, and only two structures show the combined features of an enlarged His70-Tyr99 interatomic distance (>3.5 Å) and the perpendicular Phe9 side chain conformation. The exceptions are two extraordinary wtA2/peptide structures where residues elsewhere in the peptide create a void in the peptide-binding groove that leads to the opening up of the A pocket and the unlocked state of the triad (Supplementary Fig. 4). Taken together, the comparison with the PDB structures shows that the locked A pocket corresponds to the peptide-bound state and the unlocked A pocket corresponds to the peptide-receptive state of A2.

We next asked whether binding of the methionine side chain of the GM dipeptide in the F pocket of the dsA2/GM2 structure (Fig. 3b) triggers conformational rearrangements of the F pocket side chains compared with the smaller C-terminal valine side chain of the NV9 peptide. This is indeed the case (Fig. 4d, e): to accommodate the methionine, the side chain of Tyr116 flips towards the center of the peptide-binding groove to form a hydrogen bond between the phenolic hydroxyl group and Arg97, which is also pushed in the direction of the A pocket. The realignment of Tyr116 and Arg97 affects the side chain of neighboring His114, which now forms a hydrogen bond with the side chain of Tyr116. On the $\alpha_1$ helix, the side chains of His70 and His74 change conformation to accommodate two alternate hydrogen bond networks (Fig. 4d, e). We conclude that A2 can accommodate a bulky side chain such as methionine in the F pocket by increasing the volume of the binding cavity through a coordinated rearrangement of a triad of residues, namely Arg97-His114-Tyr116.

We next compared the side chains surrounding the F pocket in the peptide-bound and peptide-free states. In the dsA2/peptide_-free-2 structure, the F pocket only contains single solvent molecules. It has expanded because the hydrogen bond between Tyr116 on the β sheet floor and His74 on the $\alpha_1$ helix is no longer present. Instead, the side chain of Tyr116 has flipped towards the imidazole side chain of neighboring His114. We propose that this

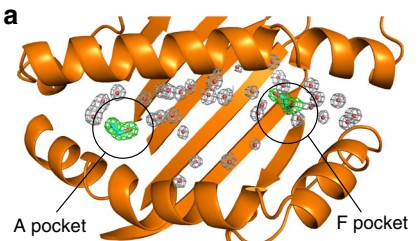

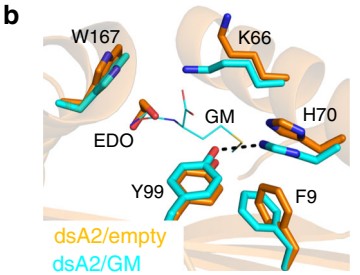

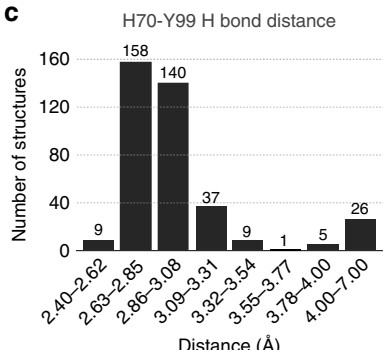

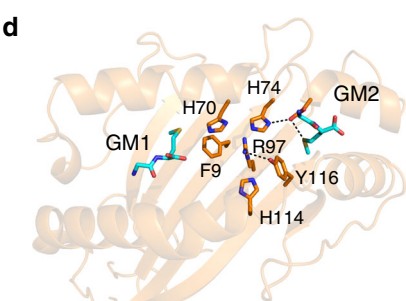

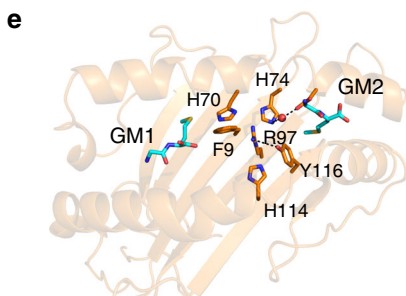

**Fig. 4 The locked and unlocked states of the A and F pockets of A2.**
**a** Ribbon diagram of the dsA2/peptide_free-1 crystal structure, showing electron density from a $2mF_o$–$DF_c$ map for the water molecules (gray) and the EDO molecules (green) in the peptide-binding groove at a contour level of 1σ. **b** Close-up of the A pocket in the locked (cyan, from the dsA2/GM structure) and the unlocked (gold, from the peptide-free structure) states. In the unlocked state, the hydrogen bond between Tyr99 and His70 is broken and the side chains of His70 and Phe9 move into the peptide-binding groove, whereas the side chain of Tyr99 moves downwards. **c** Systematic analysis of the distance distribution between the OH atom on Tyr99 and the closest nitrogen atom on the imidazole side chain of His70 for all peptide-loaded HLA-A0201 molecules observed in the PDB. The histogram shows that >90% of the deposited structures contain a hydrogen bond between His70 and Tyr99. **d**, **e** Two alternate hydrogen bond networks that are formed as a result of the presence of the methionine side chain in the F pocket are seen in each dsA2/GM₂ molecule. This leads to conformational changes in the side chains of residues His74 and Phe9.

dsA2/peptide_free-1 structure, there is an EDO molecule that sits deep inside the pocket where the methionine side chain is situated in the dsA2/GM₂ structure, but the conformation of the Arg97-His114-Tyr116 triad is the same as in the dsA2/peptide_free-2 structure. Both peptide-free dsA2 structures therefore show an unlocked F pocket through the coordinated rearrangement of the side chains of this allosteric triad, similar to the dsA2/GM₂ structure.

When comparing with the wtA2/peptide structures from the PDB database, we found that in ~25% of the deposited structures, Arg97 and Tyr116 form a hydrogen bond. Thus, a partially unlocked F pocket is quite common. Strikingly, among the 258 wtA2/peptide PDB structures examined, only one shows a fully unlocked F pocket with the formation of a hydrogen bond between His114 and Tyr116 (Supplementary Fig. 5). We conclude that like in the A pocket, the locked state of the F pocket corresponds to the peptide-bound form of A2, and the unlocked state corresponds to the peptide-receptive form, with a substantial fraction of A2/peptide structures in a partially unlocked conformation of the F pocket.

**MD simulations show allosteric effects upon peptide binding.**
To assess the conformational stability of the empty dsA2 and the dsA2/GM₂ molecules, we next performed explicit solvent MD simulations. On the time scale of 0.5 μs, no major conformational changes were observed. The backbone root-mean-square deviations (RMSD) of the α₁/α₂ peptide-binding region (residues 1–175) remained below 2 Å with respect to the crystal start structure. However, an overall slightly larger RMSD was observed for empty dsA2 (Supplementary Fig. 6a). The GM dipeptide bound to the F pocket showed greater mobility compared with the GM in the A pocket and a tendency of partial and reversible dissociation/association (Supplementary Fig. 6b). This is not unexpected given the estimated low affinity of GM binding in the mM range[11]. It also agrees with the crystal structure determination that demonstrated a larger mobility of the GM bound to the F pocket (see above). During the simulations, slightly larger fluctuations of the empty dsA2 compared with dsA2/GM₂ (Supplementary Fig. 6c) mainly in the α-helical regions of α₁/α₂ domain were observed. The modest fluctuations indicate no significant tendency of unfolding even in the empty dsA2. The presence of the GM dipeptides further reduces the fluctuations of α-helices that flank the binding groove. During the simulations of dsA2/GM₂, the His70 and Tyr99 side chains remain close in

conformation is the unlocked state of the F pocket (Fig. 5a). Interestingly, Tyr116 forms a hydrogen bond with Arg97 in a similar fashion as when GM is bound in the F pocket (Fig. 5b). The side chain of Arg97 is pushed towards the C pocket, leading to a local concentration of positive charge at the center of the peptide-binding groove (Fig. 5c). In the F pocket of the

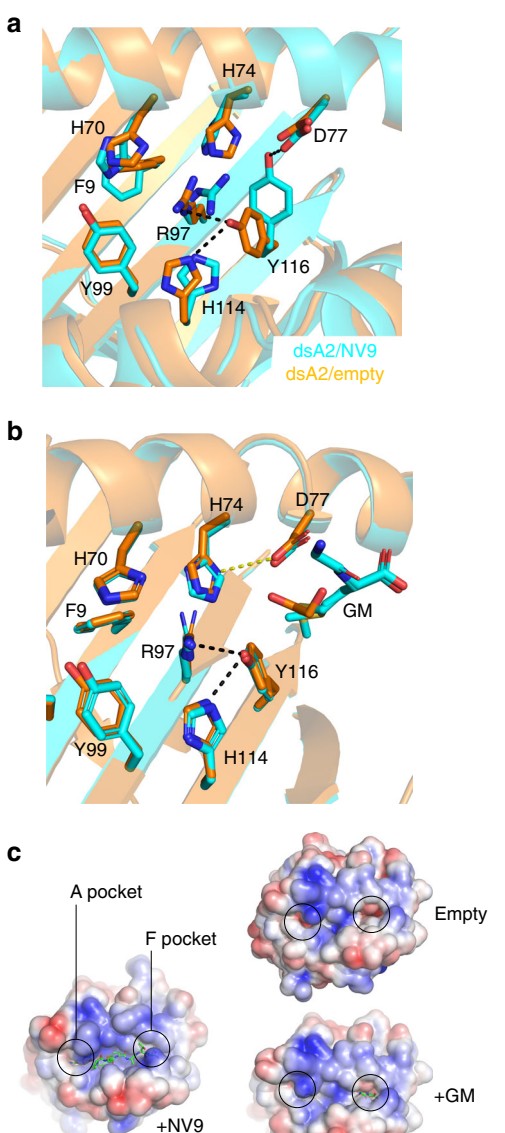

**Fig. 5 Superposition of the unlocked and locked states of the F pocket and comparison of the electrostatic surface of the peptide-free and -occupied dsA2 molecules. a** The unlocked state (derived from the dsA2/peptide_free-1 structure) is in gold, and the locked state (from the dsA2/NLVPMVATV structure) in cyan. The side chains involved in locking the F pocket are His74, Asp77, Arg97, His114, and Tyr116. The adjacent residues involved in locking the A pocket (Phe9, His70, and Tyr99) are also depicted to show their close proximity. **b** Top view of the F pocket in the unlocked state showing that the conformations of the side chains of the peptide-free dsA2 molecule coincide with the side chain conformations of one of the conformational states of dsA2/GM₂ structure, where the methionine side chain is not involved in interactions with the side chains lining the F pocket. The only difference is a hydrogen bond between His74 and Asp77 in the peptide-free dsA2 structure (yellow dotted line). **c** Electro-surface potential of the peptide-binding groove of peptide-loaded dsA2 (+NV9), dsA2/GM₂ (+GM), and peptide-free dsA2 (empty). The A and F pockets are marked to show their opening up, and the accompanying charge shifts, in the unlocked state.

partially hydrogen bonding contact (representing a locked state), whereas in the empty dsA2, larger fluctuations and conformational rearrangement were observed (Supplementary Fig. 7), transiently disrupting the His70-Tyr99 contact completely. The

result is qualitatively compatible with the unlocked state observed for the empty dsA2, which shows a disrupted His70-Tyr99 contact but less dramatic than seen during the simulations (Supplementary Fig. 7).

To gain insight into the free energy barriers of the large coordinated side chain movements that accompany the opening of the F pocket from the locked to the unlocked state, we next performed umbrella sampling (US) free energy simulations on the obtained dsA2 crystal structures. During these simulations, we probed the conformational space of the Tyr116 side chain dihedral angle ($\chi$1)[17] (Fig. 6, Supplementary Fig. 8). The *gauche* conformation of the Tyr116 side chain corresponds to the unlocked state of the F pocket (the peptide-free crystal structures), whereas the *trans* conformation corresponds to the locked state of the F pocket. Calculations on the dsA2/peptide_-free-1 structure indicated local free energy minima for both *gauche* and *trans* conformations, with the *gauche* conformation predicted to be slightly more stable. The simulations reveal a significant free energy barrier (2.5–3 kcal/mol) between *gauche* and *trans* even without any ligand bound (Fig. 6a, black curve), consistent with an allosteric conformational switch between the unlocked and the locked states of the F pocket. The free energy landscape changed significantly when we considered the peptide-bound dsA2/NV9 structure, which has a valine residue of the peptide in the F pocket (Fig. 6a, red curve). The introduction of the valine side chain leads to steric clashes between Tyr116 and neighboring residues, resulting in the destabilization of the unlocked *gauche* and the stabilization of the locked *trans* conformation.

Strikingly, when the GM is bound, exactly the opposite behavior is observed. Now, the unlocked *gauche* conformation of Tyr116 is significantly less stable owing to the influence of the large methionine side chain in the F pocket (Fig. 6a, green curve). We next monitored how in the MD simulations, the conformation of the Tyr116 side chain affects the conformation of neighboring side chains, and we detected a clear correlation (Fig. 6b). The side chain of Arg97 aligns toward the center of the peptide-binding groove for the unlocked conformation of the Tyr116 side chain, consistent with the conformations observed in the crystal structures of the empty or GM bound dsA2 structures. In accordance, His114 adopts a side chain dihedral pattern that allows transient hydrogen bonding to Tyr116. Thus, the unlocked state of the F pocket seen in the crystal structure is recapitulated in the MD simulations. When, in contrast, Tyr116 is in the locked conformation, the side chain of Arg97 is switched accordingly, and also His70 and His74 are somewhat affected. In the locked state of the F pocket, the phenolic hydroxyl group of Tyr116 makes a hydrogen bond with an oxygen atom of Asp77. In the unlocked state, however, the same hydroxyl group of Tyr116 forms a hydrogen bond with the side chain of His114, likely causing the altered side chain dihedral pattern. Indeed, Arg97 is most affected by the conformational switch of Tyr116. This seems to have a knock-on effect on His70, which is situated close to the side chain of Arg97 but on the opposite side relative to Tyr116. Hence, the simulation supports an allosteric model where the Arg97-His114-Tyr116 triad switches conformation upon engagement with a peptide from the unlocked to the locked state in a concerted fashion. All conformational changes occur towards the center of the peptide-binding groove and reshape the C, D, E, and F pockets of A2. The results of the MD simulations thus reinforce the conclusions drawn from the crystal structures and suggest a significant energy barrier between the unlocked and the locked conformational states, and a concerted conformational change of the Arg97-His114-Tyr116 triad upon peptide binding.

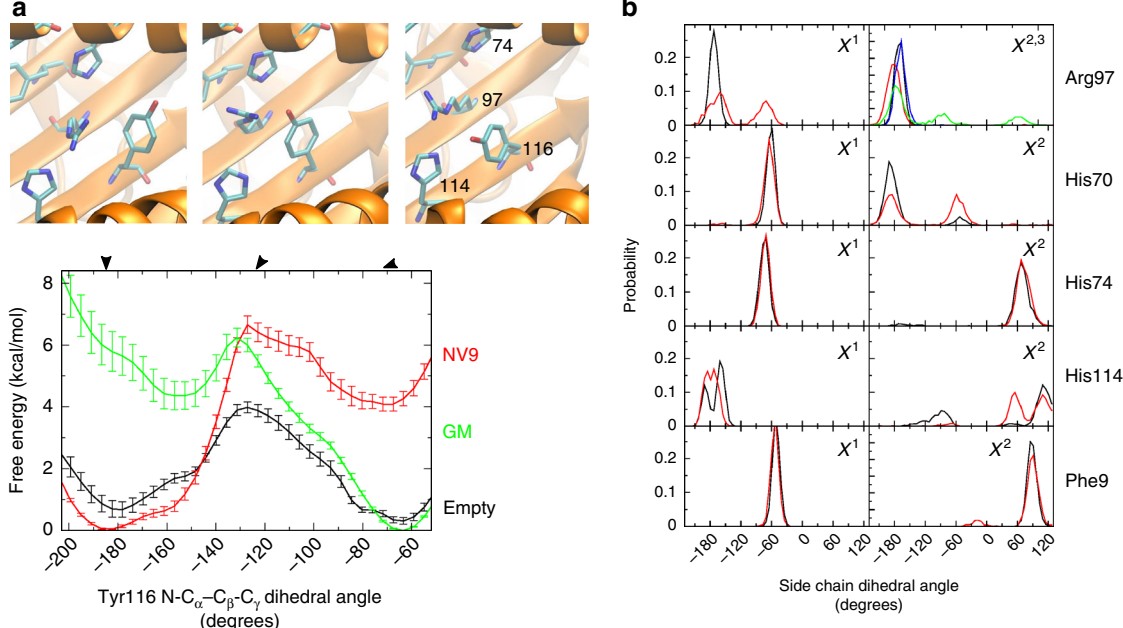

**Fig. 6 The dynamics of the transition between the unlocked and the locked states of the F pocket demonstrated by molecular dynamics simulations.**
**a** The graphs show the calculated free energy along the Tyr116 χ1 dihedral angle coordinate obtained from umbrella sampling free energy simulations (black: empty dsA2, red: dsA2/NV9, green: dsA2/GM₂). The three structure images correspond to the states along the dihedral reaction coordinate indicated by the arrows: left, Tyr116 χ1 in near-trans (i.e., locked) conformation as found in the peptide-bound state; center, Tyr116 χ1 in an energetically unfavorable transition state; right: Tyr116 χ1 in -*gauche* close to the unlocked conformation found in the peptide-free or GM bound states of dsA2.
**b** Distribution of sampled side chain dihedral angles of residues near Tyr116 in the case of Tyr116 χ1 being in the *trans* regime (locked, peptide-bound state, black curve [blue for Arg97 χ 3]) and in the case of Tyr116 χ1 being in the -*gauche* regime (unlocked state, red line [green for Arg97 χ3]).

## Discussion

The binding of a peptide to a particular MHC class I allotype (and thus its surface presentation) depends on the match of its N terminus and some of its amino-acid side chains into the binding pockets of the peptide-binding groove. Until now, the study of the dynamic processes that accompany peptide binding has been hindered by the lack of a crystal structure of the peptide-free form of class I. In this paper, we present such a structure for the disulfide-stabilized form of HLA-A*02:01, and we describe the concerted conformational changes of the amino-acid side chains lining the class I peptide-binding groove that occur when a peptide binds.

We find that both the A and F pockets contain side chain triads that undergo such concerted changes. In the A pocket, peptide or dipeptide binding switches the Phe9-His70-Tyr99 triad into a locked state that affects the B and C pockets. MD simulations on the empty dsA2 indicate that indeed the His70-Tyr99 contact is largely disrupted, whereas with bound GM in the A pocket a locked state with a more-stable His70-Tyr99 contact is preserved. In the F pocket, however, GM binding, with a methionine in the F pocket, supports the unlocked state of the Arg97-His114-Tyr116 triad that is otherwise found in the peptide-free structure. Our free energy MD simulations to induce the Tyr116 side chain transition indicate a considerable energy barrier between the locked and unlocked states in the absence of a ligand and hence support the view of a concerted conformational switch between these two states, rather than a mere conformational fluctuation. The switch of the F pocket triad also affects the shapes of the C, D, and E pockets.

The F pocket results explain why methionine is not a common C-terminal residue of A2-binding peptides. Methionine-binding switches the F pocket to the unlocked conformation, even with the peptide bound, which impacts the conformation

of His74 such that it interferes with the binding of the peptide backbone. The binding of a peptide that inserts a C-terminal methionine into the F pocket is thus predicted to be weak, which potentially explains why there are only four HLA-A2 crystal structures in the PDB that contain methionine as the C-terminal peptide residue. In all four of these structures, conformational adjustments elsewhere in the peptide compensate for the low binding energy of the C-terminal methionine (Supplementary Fig. 9).

Of all MHC class I allotypes, A2 is most thoroughly studied in structural terms. This has enabled us to conduct a thorough assessment of the changes in the conformational state of the peptide-binding groove when the peptide is removed by comparing our structures to 258 A2/peptide structures in the PDB. We were able to define two states for both the A and the F pockets: the locked state, which represents most peptide-loaded MHC structures, and the unlocked state, which is found in peptide-free class I molecules. Crystal structures and MD simulations allow us to conclude that upon binding of a full-length peptide to an empty class I molecule, the A pocket switches to the locked state in nearly all cases, whereas the F pocket remains unlocked in 25% of A2 structures with full-length peptide.

The unlocked state of the F pocket is more open to the solvent (Fig. 5b). Thus, the lower tendency of the F pocket to switch to the locked state even when high-binding peptides are bound supports the notion that peptide exchange at the F pocket can occur unassisted as shown for a number of class I allotypes[4,18]. The chaperones TAPBPR and tapasin can further destabilize the peptide-binding groove to remove peptides from the F pocket (and, in consequence, the binding groove[19]). Indeed, recent structures of MHC Class I in complex with TAPBPR[20,21] or a tapasin loop fragment[22] suggest that such a

transition state at the F pocket contributes to peptide exchange (Supplementary Fig. 10). Taken together with the literature, our data confirm the importance of the F pocket as the region of the protein where peptide binding can be modulated by external factors.

We believe that the concerted switch to unlock the F pocket that we describe here is not limited to A2 but may apply to many other allotypes. A similar movement upon the opening of the F pocket of HLA-B*44:02 has been suggested previously to contribute to chaperone-independent peptide exchange[23]. In addition, from a survey of class I structures in the PDB, we found another series of structures that illustrate the presence of an allosteric switch in the F pocket. HLA-A*24:02 (A24) binds the highly immunogenic peptide RYPLTFGWCF[24](PDB 3qzw). The C-terminal peptide residue, phenylalanine, binds into the F pocket, which contains Tyr116 in a position identical to A2. The side chain of Tyr116 locks the F pocket by forming a hydrogen bond with Asp74 on the α1 helix. Interestingly, a truncated octamer peptide with the sequence RYPLTFGW can also bind A24 in a stretched backbone conformation such that Trp8 fills the F pocket (PDB 5hgb). The bulky side chain of Trp8 then unlocks the F pocket, and Tyr116 switches conformation to form a hydrogen bond with His114 on the β sheet floor. That very same hydrogen bond is formed in A2 when the F pocket unlocks. The third member of the allosteric triad in A24 is Met97, which points upwards, out of the peptide-binding groove, when the F pocket is unlocked. These examples suggest that the concerted side chain switches involved in unlocking the F pocket may occur in many class I allotypes.

## Methods

**Production and structure determination of dsA2 molecules**. HLA-A*02:01 (Y84C) disulfide mutant (dsA2) heavy chain and human β2m light chain were expressed in *Escherichia coli* Rosetta BL21(*DE3*) *pLysS* using pET plasmid under the control of a T7 promoter. Cells were pelleted and lysed in cell lysis buffer (25% sucrose, 1 mM EDTA, 1 mM PMSF, 10 mM DTT in 50 mM Tris-Cl pH 8.0). Inclusion bodies were harvested by centrifugation, washed three times with detergent buffer (50 mM Tris pH 8.0, 150 mM NaCl, 0.5% Triton X 100, 1 mM DTT), followed by final wash with TBS (50 mM Tris pH 8.0, 150 mM NaCl, 1 mM DTT). The pellet was dissolved in solubilization buffer (6 M Guanidine HCl, 50 mM HEPES pH 6.5, 0.1 mM DTT, 0.1 mM EDTA) for 24 h at 4 °C. The solubilized protein was centrifuged at 30,000 × *g* for 30 min and the supernatant fraction was stored at −80 °C until use.

The refolding reaction was performed by diluting 1 μM of dsA2 heavy chain and 2 μM of hβ2m in a refolding buffer (100 mM Tris·Cl pH 8, 0.5 M arginine, 2 mM EDTA, 0.5 mM oxidized glutathione, 5 mM reduced glutathione, 10 mM GM (Bachem)) and incubated for 4–5 days at 4 °C with constant stirring, followed by protein concentration using 30 kDa cutoff membrane filters (Vivaflow-200; Sartorius). The concentrated protein was purified by SEC in 20 mM Tris-HCl pH 8.0, 150 mM NaCl on an AKTA system (GE healthcare) using a Hiload Superdex 200 16/600 column (GE healthcare). Peak fractions were pooled, concentrated to 15 mg/mL and further used for crystallization.

Crystals of dsA2 with GM were obtained by vapor diffusion from sitting drops containing an equal volume of protein (15 mg/mL) and a solution of 0.2 M magnesium acetate, 0.1 M sodium cacodylate pH 6.5, and 20% PEG 8000. Crystals of dsA2/GL were obtained from sitting drops containing an equal volume of protein (14 mg/mL) and a solution of 0.1 M HEPES pH 7.5, 20% PEG 10,000, 8% ethylene glycol (or EDO). Single crystals of the dsA2/GM and dsA2/GL complexes were transferred to a cryo-protectant solution containing 0.2 M magnesium acetate, 0.1 M sodium cacodylate pH 6.5, 20% PEG 8000, and 10% glycerol. To obtain peptide-free dsA2 crystals, two protocols were then used. For the dsA2/peptide_free-1 crystal, a single dsA2/GM crystal was incubated in a buffer containing 0.1 M HEPES pH 7.5, 20% PEG 10000, and 12% ethylene glycol. It was then transferred with a cryo loop into a nitrogen stream at 100 K and returned to the same buffer. Several X-ray data sets were collected to see if the dipeptide had been removed from the peptide-binding groove, and the freeze/thaw cycle had to be repeated twice to empty the groove. For the dsA2/peptide_free-2, a single dsA2/GM crystal was transferred to a buffer containing 1 M sodium chloride, 0.1 M HEPES pH 7.5, 20% PEG 10,000, and 12% ethylene glycol and incubated for 10 min. For the dsA2/GM₂ complex, a single dsA2/GM crystal was soaked in a solution containing 10 mM GM dipeptide, 0.1 M HEPES pH 7.5, 20% PEG 10,000, and 12% ethylene glycol.

Crystals were mounted and cryo-cooled to 100 K on the EMBL P13 beamline at DESY (Deutsches Elektronen-Synchrotron, Hamburg, Germany) containing a

PILATUS 6 M detector. X-ray data sets were collected with an MD3 goniometer, and statistics are presented in Supplementary Table 2. The data were processed with XDS[25] and scaled with AIMLESS[26]. Molecular replacement was performed using MOLREP[27] with the coordinates of a dsA2 molecule (PDB 6Q3K), and the structure was refined with REFMAC5[28] using the amplitude-based detwinning algorithm. Twinning fractions for the EDO-derived crystals are reported in Supplementary Table 2, where the main twinning fraction belongs to Miller indices *h, k, l* and the minor twinning fraction belongs to −*h, -k, l*. The engineered disulfide bond was manually built with Coot[29]. Molprobity[30] was used to validate the geometry, and coordinates and structure factors will be deposited to the PDB (see Table 1 for PDB codes).

**Thermal denaturation assay**. To measure thermal stability, dsA2 and wtA2 (0.1 mg/mL) were incubated for 30 min with 10 mM GM prior to performing a serial dilution (1:2) in SEC buffer (20 mM Tris pH 8, 150 mM NaCl). After 1 h equilibration, samples were used to fill two standard grade NanoDSF capillaries (Nanotemper, Munich, Germany) and loaded into a Prometheus NT.48 fluorimeter (Nanotemper) controlled by PR.ThermControl (version 2.1.2). Excitation power was pre-adjusted to obtain fluorescence readings above 2000 RFU for fluorescence emission at 330 nm (F330) and F350, and samples were heated from 20 °C to 90 °C with a rate of 1 °C/min. Data were analyzed with GraphPad Prism.

**Native MS**. In advance of native MS measurements, Vivaspin 500 centrifugal filter units (molecular weight cutoff 10 kDa; Sartorius) were used at 13,000 × *g* and 4 °C to exchange purified protein samples from 250 mM to 1 M ammonium acetate (99.99% purity; Sigma-Aldrich), pH 8.0 as buffer surrogate. For native MS experiments, the final concentration of the dsA2 protein as well as the wild-type A2 complex was 10 μM. Native MS analysis was implemented on a Q-Tof II mass spectrometer in positive electrospray ionization mode. The instrument was modified to enable high mass experiments (Waters and MS Vision)[31] Sample ions were introduced into the vacuum using homemade capillaries via a nano-electrospray ionization source in positive ion mode (source pressure: 10 mbar). Borosilicate glass tubes (inner diameter 0.68 mm, outer diameter 1.2 mm; World Precision Instruments) were pulled into closed capillaries in a two-step program using a squared box filament (2.5 mm × 2.5 mm) within a micropipette puller (P-1000, Sutter Instruments). The capillaries were then gold-coated using a sputter coater (40 mA, 200 s, tooling factor of 2.3 and end bleed vacuum of 8 × 10⁻² mbar argon; Q150R, Quorum Technologies Ltd.). Capillaries were opened directly on the sample cone of the mass spectrometer. In regular MS mode, spectra were recorded at a capillary voltage of 1.2–1.45 kV and a cone voltage of 120–150 V. Protein species with quaternary structure were assigned by MS/MS analysis. These experiments were carried out using argon as collision gas (2.0 × 10⁻² mbar). The collision voltage ranged from 10 V to 150 V. Comparability of results was ensured as MS quadrupole profiles and pusher settings were kept constant in all measurements. A spectrum of cesium iodide was recorded on the same day of the particular measurement to calibrate the data. All spectra were evaluated via MassLynx (Waters). By means of this software, peaks were assigned and the respective masses determined. The values of the shown averaged masses of the different species as well as the corresponding standard deviation result from at least three independent measurements. Also given are the average full width at half maximum of each species (Supplementary Table 1). Narrow peak widths indicate rather homogeneous samples.

**MD free energy simulations**. The crystal structures of dsA2/peptide_free-2, dsA2/GM₂, and dsA2/NV9 served as starting structures for MD simulations. All simulations were performed using the Amber18 package[32]. Proteins were solvated in octahedral boxes including explicit sodium and chloride ions (0.15 M) and explicit 4-point OPC water molecules[33] keeping a minimum distance of 10 Å between protein atoms and box boundaries. The parm14SB force field[34] was used for the proteins and peptides. The simulation systems were first energy minimized (5000 steps) followed by heating up to 300 K in steps of 100 K with position restraints on all heavy atoms of the proteins. Subsequently, positional restraints were gradually removed from an initial 25 kcal mol⁻¹ Å⁻² to 0.5 kcal mol⁻¹ Å⁻² within 0.5 ns followed by a 1 ns unrestrained equilibration at 300 K. All production simulations were performed at a temperature of 300 K and a pressure of 1 bar. The hydrogen mass repartition option of Amber was used allowing a time step of 4 fs[35]. On the empty dsA2 and dsA2/GM₂ systems unrestrained production simulations for up to 500 ns were performed. The US method was employed in order to induce dihedral transitions of the χ1 side chain dihedral of the Tyr116 residue during US simulations. A quadratic penalty potential was added to the force field to control the χ1 side chain dihedral angle (defined by the N, Ca, Cb, and Cg atoms of the Tyr116 residue) with a force constant of $k = 0.015$ kcal mol⁻¹ deg⁻². The reference dihedral in the penalty potential was varied in steps of 10° along the range of −210° to −50° (17 US windows) that covers the range of experimentally observed dihedral angle states of the Tyr116 residue in the different dsA2 structures. US simulations were performed on all three systems (empty dsA2, dsA2/GM₂, and dsA2/NV9). In order to avoid large backbone changes and dissociation of the bound ligand during the simulations weak positional restraints on the backbone Cα positions (force

constant $k = 0.25$ kcal mol$^{-1}$ Å$^{-2}$) were included (allowing full conformational freedom of all side chains). After equilibration of 2 ns in each US window, simulations were extended to 50 ns to obtain a converged dihedral distribution. The associated free energy change along the coordinate (potential-of-mean force: PMF) was obtained for each system using the weight histogram analysis method[36]. The calculated dihedral reaction coordinate auto correlation time $\tau$ in each US window was $\tau < 0.4$ ns. The US simulations were split into 10 intervals (each 5 ns » $\tau$) to calculate standard errors of the PMF. The convergence of the PMFs was further checked by calculating cumulative PMFs for the first 20 ns, 30 ns, 40 ns and the full sampling time in each US window (see Supplementary Fig. 8). In each case the calculated free energy curve converged to a stable PMF that changes little with increasing simulation time. Analysis of RMSD and root-mean-square fluctuations as well as side chain dihedral distributions of residues Phe9, His70, Arg97, His74, and His114 during the US simulations was performed using the cpptraj module of the Amber18 package.

**Reporting summary**. Further information on research design is available in the Nature Research Reporting Summary linked to this article.

## Data availability

Crystal structure coordinates and structure factors have been deposited to the Protein Data Bank under accession codes dsA2/GM (6TDO), dsA2/GL (6TDP), dsA2/GM2 (6TDQ), dsA2/peptide_free-1 (6TDR), and dsA2/peptide_free-2 (6TDS), respectively. The thermostability fluorescence data for wtA2 and dsA2 in the presence of different dipeptide concentrations are reported as the 350/330 ratio in the Source Data file for Fig. 1a, b. The outcome of the dihedral angle distribution from the MD simulations is also reported in the Source Data file for Fig. 6a, b. Other data are available from the corresponding authors upon reasonable request.

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

## Acknowledgements

We acknowledge the Sample Preparation and Characterization Facility at EMBL Hamburg and the Macromolecular P13 beamline at EMBL Hamburg/DESY. This work has been supported by iNEXT, grant number 653706, funded by the Horizon 2020 program of the European Union. The Heinrich-Pette-Institute, Leibniz Institute for Experimental Virology, is supported by the Freie und Hansestadt Hamburg and the Bundesministerium für Gesundheit (BMG). C.U. acknowledges funding from the Leibniz Association through SAW-2014-HPI-4 grant and J.L. through LFF Deligrah. S.S. acknowledges funding by the Deutsche Forschungsgemeinschaft (SP583/12-1). We acknowledge excellent technical support by Ursula Wellbrock and early contributions by Stéphane Boivin.

## Author contributions

A.R. produced protein and assisted in crystallization and X-ray data collection, M.G.A. performed thermal denaturation experiments, crystallized proteins, and assisted in X-ray data collection, J.D.K., J.L., and C.U. collected and analyzed MS data, M.A., J.H., and I.M.N. assisted in sample preparation and biophysical characterization, M.Z. performed

all MD simulations, R.M. determined the X-ray structures, R.M. and S.S. conceived the work, supervised experiments, and wrote the manuscript with input from all authors.

## Competing interests

S.S. and M.Z. are listed as inventors for a patent granted to produce empty MHC-I molecules (patent no. US9494588B2). S.S. is inventor for a published US patent application (14/370,217), describing a method for using helper ligands to allow folding of a receptor protein. S.S. and Jacobs University Bremen are partial owners of Tetramer Shop (https://tetramer-shop.com). All other authors declare no competing interests.
