## [Peer Review File · Nature Communications]

Reviewers' comments:

Reviewer #1 (Remarks to the Author):

The manuscript NCOMMS-19-26723 by Anjanappa et. al., entitled "Structures of peptide-free and partially loaded MHC class I molecules reveal mechanisms of peptide selection", report on the study of X-ray crystal structures of the peptide-free state of HLA-A*02:01, together with structures that had dipeptides bound in the A and F pockets. The author proposed that peptide binding to Major Histocompatibility Complex (MHC) class I molecules is co-determined by synergy between the binding pockets of the MHC molecule. To validate the hypothesis, they used two dipeptides, GM and GL, to examine the stability by native mass spectrometry.

This reviewer strongly agree with the claims that a thorough understanding of the selection of high-affinity peptides by class I molecules is essential for the design of peptide vaccines and for the analysis and prediction of cytotoxic T cell responses to infection and cancer. Therefore, it can be accepted for a publication in Nature Communications with the following issues being addressed.

1. The authors stated that "we did not detect the dipeptide used for refolding either free or bound to dsA2, which confirms that the dsA2/ β 2m complex is indeed empty and stable in absence of any peptide in the binding groove (Figure S1)".

--- Did the authors have tested other low affinity or no affinity dipeptides? When analysing with native mass spectrometry, any positive controls of dipeptides?

2. Page 5, "Upon ionization in the mass spectrometer, individual heavy chain or β 2m molecules are released from the complex and appear as individual peaks (Table S1)."

--- Did the authors use LC to confirm that the β 2m molecules only released after ionization?

3. We added a tenfold molar excess of NV9 to dsA2/GM and analyzed the resulting complexes by native MS (Figure 2B).

--- Did the authors perform negative control experiments? Is it possible due to the high concentration?

4. We conclude from this structure that dsA2 can accommodate two dipeptides at the same time, one in the A and one in the F pocket, and that they are bound like the N and C termini of a nonamer peptide.

--- A table is needed to elucidate all detected ions and their assignment, together with mass accuracy.

--- Is it possible to obtain the MS spectra corresponding to the co-existence of two dipeptides in a single MHC complex?

Reviewer #2 (Remarks to the Author):

Anjanappa et al. characterized peptide-free empty HLA-A2 molecules and determined their crystal structures. Based on their previous investigations, they successfully prepared a stable HLA-A2 form (dsA2) by introduction of Cys84-Cys139 disulfide bond located close to peptide (C-terminal) binding site together with addition of dipeptides, GL or GM, confirmed by native mass spectrometry. Furthermore, they also determined the crystal structures of dsA2 with GL or GM, and peptide-free form with 1,2 ethanediol (EDO), which is the first important example of peptide-free HLA/MHC molecules. The A and F pockets were identified to have two states, "locked" and

"unlocked" states. Molecular dynamics simulation supported these conformational changes and interaction network of important residues at both sites. They also discussed the peptide exchange mechanism and peptide preferences. These results provide novel insights on the functional and structural significance of peptide-free, -exchange, -loading system of HLA/MHC molecules. This paper is interesting and has potential to advance our knowledge for a field of immunology. I have some comments.

Comments

1. The SEC results of the peptide-complexed, dipeptide-complexed and peptide-free HLA molecules should be shown.
2. The authors should further perform MD simulation of the peptide-free wtA2 structure for comparison with dsA2 structures, which may provide the role of the disulfide bond stabilizing the peptide-free HLA molecule.
3. The authors should show the structural comparison between the peptide-free dsA2 structures and TAPBPR- or tapasin-complexed HLA structures. This will be helpful for understanding the discussion of peptide loading mechanism.
4. The interaction list with dipeptides or EDO of all structures will be helpful.
5. Some figures are difficult to see structural comparisons, such as Figures 3C and S3. Different coloring and/or additional figures at different angles may be helpful.

Reviewer #3 (Remarks to the Author):

The submitted manuscript presents crystal structures of peptide-free and dipeptide-loaded HLA-A*02:01. The authors engineered a disulfide bridge in the vicinity of the MHC-I binding groove, and showed its stabilising effect on the protein complex using thermal denaturation and mass spectroscopy. By comparing HLA-A*02:01 in presence and absence of peptide, the authors identified conformational changes in two residue triads, one in the A-pocket and another in the F-pocket; these two binding pockets are known to anchor the peptide to MHC-I at its N- and C-termini, respectively. The authors propose an allosteric mechanism whereby peptide binding/unbinding is accompanied by a lock/unlock conformational switch, and they study the concerted changes at the F-pocket using free energy molecular dynamics simulations.

While MHC-I variants with a stabilising disulfide bridge have been used previously in published studies by the Springer group (Hein et al. 2018, PLOS One; Hein et al. 2014, J Cell Sci), this is the first time that a peptide-free MHC-I structure has been resolved, an important achievement for our understanding of immunity at a structural level. Furthermore, the presented structures will likely become starting points for new simulation studies of peptide-free and dipeptide-loaded MHC-I. With the recent publication of a structure of the complete MHC-I peptide-loading complex and of two structures of the TAPBPR peptide editor in complex with MHC-I, the present study is timely: MHC-I protein dynamics in the context of peptide loading will likely be a point of interest in immunity research in the coming years. The authors are appropriately critical of their work and provide robust evidence. They also discuss their claims in the context of the previous literature, and take into account the high complexity of the HLA system. For instance, they discuss how their suggested allosteric mechanism could apply to other MHC-I allotypes.

The thermal denaturation and mass spectrometry experiments show very clearly the stabilising effect of the disulfide bridge on empty MHC-I, and the X-ray structures speak for themselves, having sufficient resolution to describe the conformational changes associated with peptide binding at the atomistic level. The proposed locking mechanism at the F-pocket is very clear from the presented structures. However, the mechanism describing the locking of the A-pocket is much less

convincing. The structures of the locked and unlocked states seem very close in Fig. 4B for all residues except K66, which is not part of the allosteric triad and is not discussed at all in the main text. The authors argue that the H-bond between Y99 and H70 is broken, but their description is strictly qualitative, with no distance, RMSD or dihedral angle change given. The fact that the two conformations are seen in PDB structures they reviewed only means that both conformations exist in other systems, not that they exhibit important differences that are relevant for peptide loading. Incidentally, only the F-pocket mechanism was further studied using molecular dynamics. The A-pocket mechanism was apparently not investigated, with no explanation offered.

The free energy molecular dynamics simulations unfortunately read like an afterthought. The simulations are described so tersely that they could not possibly be reproduced. For instance, no information whatsoever is given about the potential (force field, water model) or the simulation setup. The authors refer to 50 ns of umbrella sampling, but it is not clear if that is total or per umbrella window, or if any equilibration was performed. More importantly, the number of windows is not given. There seems to be no error estimate or any check for statistical validity, such as assessing sufficient conformational space sampling or checking for correlated samples. The Mobley group published excellent guidelines for the analysis of free energy calculations that could be of use to the authors (Klimovich et al. 2015, *J Comput Aided Mol Des*). Standard (unbiased) MD simulations and their results should also be performed and briefly discussed, if only to assess the stability of the system. For instance, do the dipeptides remain in the binding groove during μ s-timescale simulations? The low quality of the given simulation protocols contrasts with the more detailed information given for crystallisation or fluorescence measurements. The free energy simulations section should be either rewritten entirely and augmented or removed from the manuscript; it certainly cannot be published in its current form.

In conclusion, I recommend to invite the authors to revise their manuscript to address the specific concerns I expressed above before making a final decision.

Minor points:

Fig. 1A: The plots seem to extend beyond their data range near 20°C and 90°C, leading to flat lines that are probably artefacts.

Fig. 4C: H70 is incorrectly shown in orange for both the locked and unlocked states.

Fig. 3-6: These figures could be improved by adding colour keys directly on the figures rather than describing colours in the caption only. This would also be helpful for colour-blind readers.

Olivier Fissette
Centre for Theoretical Chemistry
Ruhr-University Bochum, Germany

Reviewers' comments:

We thank the reviewers for the positive assessment of the manuscript and the constructive criticism. Please find below a point-by-point discussion of the issues raised by the reviewers.

Reviewer #1 (Remarks to the Author):

*The manuscript NCOMMS-19-26723 by Anjanappa et. al., entitled “Structures of peptide-free and partially loaded MHC class I molecules reveal mechanisms of peptide selection”, report on the study of X-ray crystal structures of the peptide-free state of HLA-A*02:01, together with structures that had dipeptides bound in the A and F pockets. The author proposed that peptide binding to Major Histocompatibility Complex (MHC) class I molecules is co-determined by synergy between the binding pockets of the MHC molecule. To validate the hypothesis, they used two dipeptides, GM and GL, to examine the stability by native mass spectrometry.*

This reviewer strongly agree with the claims that a thorough understanding of the selection of high-affinity peptides by class I molecules is essential for the design of peptide vaccines and for the analysis and prediction of cytotoxic T cell responses to infection and cancer. Therefore, it can be accepted for a publication in Nature Communications with the following issues being addressed.

1. The authors stated that “we did not detect the dipeptide used for refolding either free or bound to dsA2, which confirms that the dsA2/ 2m complex is indeed empty and stable in absence of any peptide in the binding groove (Figure S1)”.

--- Did the authors have tested other low affinity or no affinity dipeptides? When analysing with native mass spectrometry, any positive controls of dipeptides?

Dipeptides have a much lower binding affinity than the nonapeptides resulting in low dissociation thresholds. Only non-significant signals of dsA2 complex with dipeptides remain at sufficient resolution. This trend also holds for longer tri- and tetrapeptides (data not shown), which hampers the use of positive or negative controls. Therefore, we performed an indirect readout and adapted the main text (pages 5-6). In figure 2A, we added the low m/z range to show the absence of dipeptide GM from refolding of dsA2 after wash out. Even though GM is depleted, we still observe the same intact dsA2 complex seen in presence of the peptide, which is not the case for wtA2. We hence regard wtA2 as control for dipeptide binding/removal.

2. Page 5, “Upon ionization in the mass spectrometer, individual heavy chain or 2m molecules are released from the complex and appear as individual peaks (Table S1).”

--- Did the authors use LC to confirm that the 2m molecules only released after ionization?

There was no need to perform LC-MS experiments, because we only observe the intact dsA2/ 2m complex at low collision voltages (up to approx. 25V) without any dissociation of 2m. The complex dissociates only at higher activation, and we are then able to observe the free 2m and heavy chain respectively (shown in the supplement previously as Figure S1, now Figure S2). This is different though for the

wtA2 complex in absence of dipeptide, where dissociation of γ_2m occurs in solution as indicated by the low charge states. To better explain the process, we have rephrased it on p. 5:

“Upon ionization in the mass spectrometer, individual heavy chain or γ_2m molecules are released from the complex and appear as individual peaks”

has been replaced by:

“Upon activation in the mass spectrometer, individual heavy chain or γ_2m molecules are released from the complex and appear as individual peaks”

3. We added a tenfold molar excess of NV9 to dsA2/GM and analyzed the resulting complexes by native MS (Figure 2B).

--- Did the authors perform negative control experiments? Is it possible due to the high concentration?

We have now performed concentration-dependent experiments with low-affinity peptides (YF9, dipeptides) and a high affinity nonapeptide (NV9). At ten-fold molar excess of dsA2, we do see NV9 present, whereas the low affinity peptide YF9 is absent (now shown in Figure 2C). Moreover, the NV9 complex is much more resistant to gas phase dissociation upon increase of collision voltage, in contrast to YF9. This is now clarified in the main text (page 6).

4. We conclude from this structure that dsA2 can accommodate two dipeptides at the same time, one in the A and one in the F pocket, and that they are bound like the N and C termini of a nonamer peptide.

--- A table is needed to elucidate all detected ions and their assignment, together with mass accuracy.

The Table S1 was already included that shows the assignments and mass accuracies.

--- Is it possible to obtain the MS spectra corresponding to the co-existence of two dipeptides in a single MHC complex?

We have not been able to obtain spectra for two dipeptides in a single MHC complex with the currently used dipeptides (GL and GM), which is related to the low affinity and small mass increase requiring activation beyond the dissociation threshold for sufficient resolution (see above). To test other dipeptides is beyond the scope of this paper and would likely provide similar results.

Reviewer #2 (Remarks to the Author):

Anjanappa et al. characterized peptide-free empty HLA-A2 molecules and determined their crystal structures. Based on their previous investigations, they successfully prepared a stable HLA-A2 form (dsA2) by introduction of Cys84-Cys139 disulfide bond located close to peptide (C-terminal) binding site together with addition of dipeptides, GL or GM, confirmed by native mass spectroscopy. Furthermore, they

also determined the crystal structures of dsA2 with GL or GM, and peptide-free form with 1,2 ethanediol (EDO), which is the first important example of peptide-free HLA/MHC molecules. The A and F pockets were identified to have two states, “locked” and “unlocked” states. Molecular dynamics simulation supported these conformational changes and interaction network of important residues at both sites. They also discussed the peptide exchange mechanism and peptide preferences. These results provide novel insights on the functional and structural significance of peptide-free,-exchange, -loading system of HLA/MHC molecules. This paper is interesting and has potential to advance our knowledge for a field of immunology. I have some comments.

Comments

1. The SEC results of the peptide-complexed, dipeptide-complexed and peptide-free HLA molecules should be shown.

We have now included the SEC profiles as requested in Figure S1.

2. The authors should further perform MD simulation of the peptide-free wtA2 structure for comparison with dsA2 structures, which may provide the role of the disulfide bond stabilizing the peptide-free HLA molecule.

We have conducted additional comparative MD simulations on the empty dsA2 and dsA2/GM₂ complexes to support the locked and unlocked states at the A and F pockets as proposed in the manuscript (see our response to Reviewer 3). The complex question how the disulfide bridge in dsA2 stabilizes the empty structure goes beyond the focus of the current study and will be subject of future studies.

3. The authors should show the structural comparison between the peptide-free dsA2 structures and TAPBPR- or tapasin-complexed HLA structures. This will be helpful for understanding the discussion of peptide loading mechanism.

We have now included a figure (Figure S10) with a structural comparison of the murine TAPBPR/Db complex structure and the human dipeptide-occupied and empty dsA2 structures, to show that the F pocket plays a crucial role in the peptide loading mechanism aided by the tapasin and/or TAPBPR chaperones.

4. The interaction list with dipeptides or EDO of all structures will be helpful.

We have now included a table (Table S3) that lists all the interactions for the dipeptides, glycerol and EDO molecules that occupy the peptide binding groove.

5. Some figures are difficult to see structural comparisons, such as Figures 3C and S3. Different coloring and/or additional figures at different angles may be helpful.

We have added a different orientation for Figure 3C (see new panel 3D) and have highlighted the different orientation of the EDO molecules in Figure S3 to better facilitate structural comparisons. We have also added a colour key to the structure figures that show overlapping structures, as suggested by Reviewer #3.

Reviewer #3 (Remarks to the Author):

*The submitted manuscript presents crystal structures of peptide-free and dipeptide-loaded HLA-A*02:01. The authors engineered a disulfide bridge in the vicinity of the MHC-I binding groove, and showed its stabilising effect on the protein complex using thermal denaturation and mass spectroscopy. By comparing HLA-A*02:01 in presence and absence of peptide, the authors identified conformational changes in two residue triads, one in the A-pocket and another in the F-pocket; these two binding pockets are known to anchor the peptide to MHC-I at its N- and C-termini, respectively. The authors propose an allosteric mechanism whereby peptide binding/unbinding is accompanied by a lock/unlock conformational switch, and they study the concerted changes at the F-pocket using free energy molecular dynamics simulations.*

While MHC-I variants with a stabilising disulfide bridge have been used previously in published studies by the Springer group (Hein et al. 2018, PLOS One; Hein et al. 2014, J Cell Sci), this is the first time that a peptide-free MHC-I structure has been resolved, an important achievement for our understanding of immunity at a structural level. Furthermore, the presented structures will likely become starting points for new simulation studies of peptide-free and dipeptide-loaded MHC-I. With the recent publication of a structure of the complete MHC-I peptide-loading complex and of two structures of the TAPBPR peptide editor in complex with MHC-I, the present study is timely: MHC-I protein dynamics in the context of peptide loading will likely be a point of interest in immunity research in the coming years. The authors are appropriately critical of their work and provide robust evidence. They also discuss their claims in the context of the previous literature, and take into account the high complexity of the HLA system. For instance, they discuss how their suggested allosteric mechanism could apply to other MHC-I allotypes.

The thermal denaturation and mass spectrometry experiments show very clearly the stabilising effect of the disulfide bridge on empty MHC-I, and the X-ray structures speak for themselves, having sufficient resolution to describe the conformational changes associated with peptide binding at the atomistic level. The proposed locking mechanism at the F-pocket is very clear from the presented structures. However, the mechanism describing the locking of the A-pocket is much less convincing. The structures of the locked and unlocked states seem very close in Fig. 4B for all residues except K66, which is not part of the allosteric triad and is not discussed at all in the main text. The authors argue that the H-bond between Y99 and H70 is broken, but their description is strictly qualitative, with no distance, RMSD or dihedral angle change given. The fact that the two conformations are seen in PDB structures they reviewed only means that both conformations exist in other systems, not that they exhibit important differences that are relevant for peptide loading.

We agree with the reviewer that the analysis of the A pocket was incomplete. Inspired by his comments, we have now systematically verified the distance between the OH atom on Tyr99 and the nitrogen atom on the imidazole side chain of His70 that is closest, and we present a histogram in Figure 4C that shows the distance distribution for all peptide-loaded HLA-A0201 molecules observed in the PDB. The histogram shows that >90 % of the deposited structures contain a hydrogen bond between His70 and Tyr99. We also verified again the side chain conformation of Phe9, and can confirm that out of 258 PDB entries with peptide loaded wtA2 structures, only three structures contain a Phe9 side chain conformation that coincides with that of the

empty dsA2 structures. This is not a surprise, because the Phe9 side chain clearly moves into the space vacated by the peptide in the peptide-free dsA2 structures. We show the peptide conformation for the two structures deposited in the PDB that show the “empty” Phe9 conformation combined with a disrupted hydrogen bond in Figure S5.

We then also had a closer look at the two empty dsA2 structures and realized there is a notable difference between the “annealed” structure and the “high salt” structure. In the annealed structure, both dsA2 molecules in the asymmetric unit show a clear disruption of the hydrogen bond between His70 and Tyr99 (closest distance is 4.30 Å). In the high salt structure, there is an additional EDO molecule in the A pocket that assumes two different positions between the two dsA2 molecules in the asymmetric unit (already described in what is now Figure S4). Both EDO molecules form a hydrogen bond with Tyr99, affecting the His70-Tyr99 hydrogen bond distance (which is borderline at 3.52 and 3.40 Å respectively). Here, the EDO molecule seems to mimic the role of the carbonyl group of the peptide that normally sits in this position. When these observations are aligned with the conformations for the third member of the allosteric triad (Phe9), it becomes clear that the annealed dsA2 structure presents a truly unlocked pocket (both Phe9 conformations are perpendicular to the most common Phe conformation assumed by 99 % of the pMHC structures). In the high salt structure, Phe9 is perpendicular in one dsA2 molecule, but it assumes both conformations in the dsA2 molecule that also contains a true but very weak hydrogen bond (3.40 Å) between His70 and Tyr99. We have rewritten the paragraph that describes the open A pocket to reflect these observations (pages 9-10).

Incidentally, only the F-pocket mechanism was further studied using molecular dynamics. The A-pocket mechanism was apparently not investigated, with no explanation offered.

We have now also performed MD simulations to verify how the distance between the imidazole ring of His70 and the Tyr99 side chain is affected by the presence of a GM dipeptide in the A pocket. Indeed, we observe that the presence of the GM peptide stabilizes the His70-Tyr99 hydrogen bond, whereas the absence of peptide leads to fluctuations and disruption of this hydrogen bond (see the additional paragraph on page 13 and the newly added Figure S7) supporting the observation seen in the crystal structures.

The free energy molecular dynamics simulations unfortunately read like an afterthought. The simulations are described so tersely that they could not possibly be reproduced. For instance, no information whatsoever is given about the potential (force field, water model) or the simulation setup. The authors refer to 50 ns of umbrella sampling, but it is not clear if that is total or per umbrella window, or if any equilibration was performed. More importantly, the number of windows is not given. There seems to be no error estimate or any check for statistical validity, such as assessing sufficient conformational space sampling or checking for correlated samples. The Mobley group published excellent guidelines for the analysis of free energy calculations that could be of use to the authors (Klimovich et al. 2015, J

Comput Aided Mol Des). Standard (unbiased) MD simulations and their results should also be performed and briefly discussed, if only to assess the stability of the system.

For instance, do the dipeptides remain in the binding groove during μ s-timescale simulations? The low quality of the given simulation protocols contrasts with the more detailed information given for crystallisation or fluorescence measurements. The free energy simulations section should be either rewritten entirely and augmented or removed from the manuscript; it certainly cannot be published in its current form.

In conclusion, I recommend to invite the authors to revise their manuscript to address the specific concerns I expressed above before making a final decision.

We agree with the reviewer that the description of the simulation part and also the methods section was incomplete. We have now extended the description of the simulation methodology (see Material and Methods, pages 22-23). It now includes the force field, water, and simulation setup. The Umbrella sampling was performed in steps of 10° in the range of the dihedral reference angle of -210° to -50° (17 windows). The sampling was performed for each US window for 50 ns. The information and error estimation as well as convergence estimation is given in the revised Methods section (and Supporting Information). As suggested by the reviewer, we also performed 0.5 μ s unrestrained MD-simulations on the empty dsA2 and dsA2/GM₂ complex. The simulations indicate that the GM-binding in the F pocket is significantly more mobile than in the A pocket (compatible with the X-ray result) and allows us also to extract results on the suggested A pocket locking and unlocking. The Results section on the simulations was rewritten and extended in the main manuscript (pages 12-14 and additional supplementary Figures S7, S8 and S9). The suggestions by the reviewer led to additional and new interesting results on the locking mechanism. It is planned to study the systems in the future more extensively using comparative simulations of the dsA2 and HLA-A2 systems (which goes beyond the focus of the current manuscript).

Minor points:

Fig. 1A: The plots seem to extend beyond their data range near 20°C and 90°C, leading to flat lines that are probably artefacts.

We have modified Fig1A to address this issue.

Fig. 4C: H70 is incorrectly shown in orange for both the locked and unlocked states.

We have changed to orientation of the Figure slightly to remove the overlap between F9 and H70, which gave a false impression that they provided two different conformations of H70.

Fig. 3-6: These figures could be improved by adding colour keys directly on the figures rather than describing colours in the caption only. This would also be helpful for colour-blind readers.

We thank the reviewer for this suggestion, and have now added a colour key to the structure figures.

REVIEWERS' COMMENTS:

Reviewer #1 (Remarks to the Author):

The revised manuscript has addressed this reviewer's concerns. Therefore, I would like to recommend it as a publication in Nat. Communications.

Reviewer #3 (Remarks to the Author):

The revised manuscript the authors propose addresses the issues I pointed out in the original version. In particular, the molecular dynamics protocols are adequately described, and the authors now also report the results of unbiased MD simulations, with interesting new insights. I recommend that the revised manuscript be accepted for publication in its current state.

Minor issues:

p. 4, typo: "dsA2]" should be "dsA2"

p. 10: "3.52 Angstrom" should be "3.52 Å" for consistency

p. 10: "less than 10 % of these structures do not have" should be reworded to "more than 90 % of these structures have"

Fig. 1: I can no longer find the arrows mentioned in the legend. The "x" is missing for powers of 10 in panel B, vertical axis. The legend describes panels A (top and bottom) and B, but the figure uses panels A, B and C instead. The corresponding system (dsA2, wtA2) should be shown on panels A and B directly, not only stated in the legend.

--

Olivier Fiset
Centre for Theoretical Chemistry
Ruhr-University Bochum

Please find below a point-by-point response to the reviewer comments, received upon resubmission of manuscript NCOMMS-19-26723A

REVIEWERS' COMMENTS:

Reviewer #1 (Remarks to the Author):

The revised manuscript has addressed this reviewer's concerns. Therefore, I would like to recommend it as a publication in Nat. Communications.

Reviewer #3 (Remarks to the Author):

The revised manuscript the authors propose addresses the issues I pointed out in the original version. In particular, the molecular dynamics protocols are adequately described, and the authors now also report the results of unbiased MD simulations, with interesting new insights. I recommend that the revised manuscript be accepted for publication in its current state.

Minor issues:

p. 4, typo: "dsA2]" should be "dsA2"

p. 10: "3.52 Angstrom" should be "3.52 Å" for consistency

p. 10: "less than 10 % of these structures do not have" should be reworded to "more than 90 % of these structures have"

Fig. 1: I can no longer find the arrows mentioned in the legend. The "x" is missing for powers of 10 in panel B, vertical axis. The legend describes panels A (top and bottom) and B, but the figure uses panels A, B and C instead. The corresponding system (dsA2, wtA2) should be shown on panels A and B directly, not only stated in the legend.

--

Olivier Fiset
Centre for Theoretical Chemistry
Ruhr-University Bochum

Response: We thank the reviewers for the constructive comments. We have modified Figure 1, fixed the typos and made the changes suggested by Reviewer #3 in the revised manuscript.